# One-Step Multifunctionalization of Flax Fabrics for Simultaneous Flame-Retardant and Hydro-Oleophobic Properties Using Radiation-Induced Graft Polymerization

**DOI:** 10.3390/polym15092169

**Published:** 2023-05-02

**Authors:** Jamila Taibi, Sophie Rouif, Bruno Améduri, Rodolphe Sonnier, Belkacem Otazaghine

**Affiliations:** 1Polymers Composites and Hybrids (PCH), IMT Mines Ales, 30319 Ales, France; jamila.taibi@mines-ales.fr (J.T.);; 2Ionisos SAS, 13 Chemin du Pontet, 69380 Civrieux-d’Azergues, France; 3ICGM, University of Montpellier, CNRS, ENSCM, 34095 Montpellier, France

**Keywords:** flame retardancy, flax fabrics, fluorinated monomer, hydrophobicity, oleophobicity, phosphonated monomer, radiografting

## Abstract

This study concerns the one-step radiografting of flax fabrics with phosphonated and fluorinated polymer chains using (meth)acrylic monomers: dimethyl(methacryloxy)methyl phosphonate (MAPC1), 2-(perfluorobutyl)ethyl methacrylate (M4), 1H,1H,2H,2H-perfluorooctyl acrylate (AC6) and 1H,1H,2H,2H-perfluorodecyl methacrylate (M8). The multifunctionalization of flax fabrics using a pre-irradiation procedure at 20 and 100 kGy allows simultaneously providing them with flame retardancy and hydro- and oleophobicity properties. The successful grafting of flax fibers is first confirmed by FTIR spectroscopy. The morphology of the treated fabrics, the regioselectivity of grafting and the distribution of the fluorine and phosphorus elements are assessed by scanning electron microscopy (SEM) coupled with energy-dispersive X-ray spectroscopy (SEM-EDX). The flame retardancy is evaluated using pyrolysis combustion flow calorimetry (PCFC) and cone calorimetry. The hydro- and oleophobicity and water repellency of the treated fabrics is established by contact angle and sliding angle measurements, respectively. The grafting treatment of flax irradiated at 100 KGy, using M8 and MAPC1 monomers (50:50) for 24 h, allows achieving fluorine and phosphorus contents of 8.04 wt% and 0.77 wt%, respectively. The modified fabrics display excellent hydro-oleophobic and flame-retardant properties with water and diiodomethane contact angles of 151° and 131°, respectively, and a large decrease in peak of heat release rate (pHRR) compared to pristine flax (from 230 W/g to 53 W/g). Relevant results are also obtained for M4 and AC6 monomers in combination with MAPC1. For the flame retardancy feature, the presence of fluorinated groups does not disturb the effect of phosphorus.

## 1. Introduction

In recent years, much attention has been paid to surface modification methods for the production of textiles with novel performances such as superhydrophobic, flame retardant, antibacterial, anti-ultraviolet features and oil–water separation [1,2,3,4,5,6,7]. Therefore, the development of functional textiles concerns much research, including flame retardancy, hydro- and oleophobicity, and smart textiles, regarded as key topics that attract much attention. Superhydrophobic surfaces have been inspired by lotus leaves, with a water contact angle higher than 150° and an ultra-low sliding angle (less than 10°). Indeed, the surface of lotus leaves displays self-cleaning and anti-contamination properties due to the presence of micro- and nanostructures that increase the roughness and reduce the droplet adhesion [8,9]. Other treatments have also been reported such as plasma etching [10]. In addition, for safety reasons, flame-retardant fabrics are relevant by introducing phosphorus flame retardants [5,6,7,11].

Nowadays, synthetic fibers are the most used items in the textile industry for various applications. However, it is worth using natural fibers due to their low ecological impact and their renewable character [12,13]. Several methods have been applied to improve the properties of natural textiles, such as plasma treatment [14,15,16], sol–gel [17,18,19] and radiation-induced graft polymerization [20,21,22]. The latter technique was used to modify natural fibers and could provide flame-retardant (FR) [5,6,7] or hydro-oleophobic properties [20,21,22] by introducing macromolecules containing phosphorus or fluorinated groups, respectively. Phosphorus-containing synthons and macromolecules have widely been used in many applications with natural fibers because of their useful capacity to improve their flame retardancy [23,24,25]. Indeed, significant improvements against fire have been reported after the modification of natural fabrics by radiografting phosphonated monomers [5,6,7,26]. Sonnier et al. [5] studied the grafting of dimethylvinyl phosphonate (MVP) and dimethyl(methacryloxy)methyl phosphonate (MAPC1) onto flax fibers by simultaneous radiografting. The grafted phosphorus content was evidenced depending on various parameters such as the applied irradiation dose and the FR monomer concentration. The grafting results indicated that for a monomer concentration of 10 wt% in the impregnation THF solution and an irradiation dose of 100 kGy, phosphorus contents of ca. 4 and 0.34 wt% were achieved for MVP and MAPC1, respectively. The authors showed that the flammability of the treated fabrics was mainly controlled by the grafted phosphorus content, which promotes the formation of a stable char. Self-extinguishing fabrics were prepared with 1.5 wt% of phosphorus only. Hajj et al. [7] reported the reactivity under electron beam irradiation and the grafting efficiency on flax fibers of different phosphorus monomers: (acryloyloxy)methyl phosphonic acid, (methacryloyloxy)methyl phosphonic acid, MAPC1, allyl phosphonic acid, vinyl phosphonic acid and MVP. These authors observed that the increase in the absorbed dose and the monomer concentration in impregnation water solution resulted in higher phosphorus grafted contents. In addition, the methacrylic monomers displayed a stronger grafting efficiency compared to the other studied monomers. Self-extinguishing fabrics were obtained when fabrics irradiated at 50 kGy were modified with MVP and MAPC1 at 10 wt% in water with a char yield between 17 and 24 wt%.

In a previous work [27], the same group also grafted MAPC1 in water onto flax fibers irradiated at various doses ranging from 5 to 100 kGy using the pre-irradiation method. The irradiation dose, MAPC1 concentration, reaction time and temperature have a direct impact on the grafting efficiency and allow obtaining high phosphorus contents (a maximum of ca. 7 wt%) in comparison with the simultaneous method (about 2 wt%). The thermal stability and the flammability of the treated flax fibers were mainly controlled by the grafted phosphorus content. Moreover, irradiation of flax fibers at a low dose of 10 kGy and treatment with 10 wt% MAPC1 at 80 °C for 30 min (less degrading conditions) resulted in self-extinguishing fabrics with a phosphorus content of 1.4 wt% and a char content of about 19 wt%.

As a matter of fact, fluoropolymers have also attracted the interest of the academic and industrial scientific communities because of their excellent properties, including high heat, UV and aging resistance, outstanding solvent and chemical stability and low surface energy [28,29]. Per- and polyfluoroalkyl substances (PFAS) with alkyl chains containing six or more carbons (C_n_F_2n+1_ with n > 6) have a strong hydrophobic and oleophobic character [17,20,30,31,32,33,34,35]. However, these low-molar-mass substances are known to be bioaccumulative, persistent, toxic, water soluble and thus mobile, and severe regulations aim at restricting them [33,34,35]. Due to new regulations limiting or prohibiting their use, new molecules or macromolecules with shorter fluoroalkyl groups (C_n_F_2n+1_ with n ≤ 5) must be used, or fluorine-free solutions must be developed. Deng et al. [21] grafted a fluorinated acrylate monomer, 1H,1H,2H,2H-nonafluorohexyl-1-acrylate (F4), onto cotton fibers by simultaneous radiografting. The authors reported that the hydrophobicity of cotton-F4 depended on the degree of grafting (DG), which for values higher than 10 wt%, led to water contact angles above 150°. On the other hand, for cotton grafted-F4 with DG values lower than 10 wt%, the hydrophobicity of treated samples was unstable, and the water droplet rapidly spread out within 1 min. A previous work describes the grafting in methanol of different fluorinated (meth)acrylic monomers such as 1,1,1,3,3,3-hexafluoroisopropyl methacrylate (M2), 2-(perfluorobutyl)ethyl methacrylate (M4), 1H,1H,2H,2H-perfluorooctyl methacrylate (M6), 1H,1H,2H,2H-perfluorooctyl acrylate (AC6) and 1H,1H,2H,2H-perfluorodecyl methacrylate (M8) to improve the hydro- and oleophobicity of flax fabrics using the pre-irradiation method [36]. Grafting of P(M4), P(M6), P(AC6) and P(M8) onto flax fabrics led to highly hydrophobic and oleophobic characters even at a low fluorine content of 0.10 wt%, and it was also evidenced that the grafted fluorine content is the only factor that controls both characteristics. Superhydrophobic (150°) fabrics were produced in the case of M8 with the formation of spherical particles corresponding to P(M8) on the surface of the fibers. High fluorine levels between 0.4 and 13.8 wt% were achieved for this monomer compared to other fluorinated monomers.

Therefore, the combination of hydro-oleophobic and flame-retardant properties by the multifunctionalization of natural fibers using radiation-induced graft polymerization is an innovative topic, and to our knowledge, no article has been reported yet.

Hence, the objective of the present work deals with the development of a one-step procedure for the multigrafting of flax using a pre-irradiation procedure to prepare multifunctional fabrics, which are both flame-retardant and hydro-oleophobic. MAPC1 was combined with M4, AC6 or M8 fluorinated comonomers for the radiografting of flax fabrics irradiated at 20 and 100 kGy. The modified fabrics were then characterized to evaluate the grafting rate of the phosphonated and fluorinated comonomers. Finally, the hydro- and oleophobic properties, as well as the fire behavior of the modified fabrics, were assessed.

## 2. Materials and Methods

### 2.1. Materials

Flax fabrics (200 g/m^2^) were provided by Hexcel (Roussillon, France). Their chemical composition was determined by solvent extraction as 81 wt% of cellulose, 13 wt% of hemicelluloses and 2.7 wt% of lignin.

Dimethyl(methacryloyloxy)methyl phosphonate (MAPC1) was purchased from Specific Polymers (Castries, France) and used as received without any purification. 2-(perfluorobutyl)ethyl methacrylate (95%, M4); 1H,1H,2H,2H-perfluorooctyl acrylate (95%, AC6) and 1H,1H,2H, 2H-perfluorodecyl methacrylate (97%, M8) (Figure 1) were purchased from ABCR—Gmbh (Karlsruhe, Germany) and distilled under vacuum before use. Values of molar mass, fluorine and phosphorus contents of the different monomers used are presented in Appendix A, Appendix A. Tetrahydrofuran (THF), methanol and methyl ethyl ketone (MEK) were purchased from Fisher Scientific (Illkirch-Graffenstaden, France).

### 2.2. Grafting Process

In the first step, flax fabrics were irradiated in air, at room temperature, under e-beam radiation (energy 9.8 MeV, power 34 kW) at doses of 20 and 100 kGy performed by Ionisos SA (Chaumesnil, France). After irradiation, fabrics were immediately cold stored (−18 °C) to preserve the generated free radicals and/or peroxides. In a second step, an impregnation solution was prepared containing 10 wt% of a mixture of fluorinated and phosphonated monomers with different molar ratios (noted F/P) and 90 wt% of methanol. The mixture was placed under nitrogen bubbling for 15 min to remove oxygen from the reaction medium. Fabric samples irradiated at 20 or 100 kGy were added to the reaction solution and kept at 65 °C for 24 h. The final step is the washing of the treated fabrics three times with THF and three times with 2-butanone (MEK) at room temperature for P(M4) and at 60 °C for P(AC6) and P(M8) to remove unreacted monomers and free fluorinated polymer chains, which were not covalently bonded to the flax structure. Finally, the treated fabrics were dried at 60 °C for 24 h and stored in a desiccator (Figure 2).

### 2.3. Measurements

#### 2.3.1. Fourier Transform Infrared Spectroscopy (FTIR)

Fourier transform infrared spectra were recorded with a Bruker VERTEX 70 spectrometer (Metrohm, Ales, France) used in attenuated total reflectance mode, by performing 32 scans between 400 and 4000 cm^−1^ with a resolution of ±2 cm^−1^.

#### 2.3.2. Scanning Electron Microscopy SEM

The fiber section of flax fabrics was analyzed using a scanning electron microscope (FEI Quanta 200) (Thermo Fisher, Ales, France). After being cut with a single-edge blade, the samples were placed on a vertical sample holder under high vacuum at a voltage of 12.5 kV and a working distance of 10 mm. To locate the presence of the fluorine and phosphorus elements in the fiber section, SEM analysis was coupled with energy-dispersive X-ray spectroscopy (EDX) (Oxford INCA Energy system, Saclay, France).

#### 2.3.3. Measurement of Phosphorus and Fluorine Contents

The grafted phosphorus and fluorine contents were determined by a multistep calculation procedure according to Appendix A, as explained below.

##### Phosphorus Content

aInductively coupled plasma atomic emission spectroscopy

Inductively coupled plasma atomic emission spectrometry (ICP-AES) is a destructive technique used to determine the elemental composition of a material. The samples underwent a preliminary mineralization step before analysis. For this, 50 mg of flax fiber was mixed with 1 mL of nitric acid (63%) and 2 mL of sulfuric acid (98%) in a Teflon^®^ container. The mixture was heated by microwaves with power ranging between 400 and 700 W following an appropriate cycle. After cooling, the mineralized solutions were then diluted with demineralized water to 50 mL before being analyzed by ICP-AES. During this step, the vaporized solution passes into the plasma chamber at 6000 °C, and the excited atoms emit spectra specific to each element. The intensity of the peak of the phosphorus element was converted into a mass percentage using a calibration curve. Each sample was analyzed twice for the reproducibility of measurements.

b.X-ray fluorescence (XRF)

Phosphorus content was determined by X-ray fluorescence by bombarding the material with X-rays. The irradiation caused a secondary X-ray emission characteristic of the elements present in the samples. An Oxford XMET 5100 X-ray fluorescence instrument (Oxford Instruments, Ales, France) was used to determine the phosphorus content in the treated flax fabrics. The samples were fixed on a flat polymer-based substrate containing no trace of phosphorus. This substrate was used to flatten the fabrics to reduce instrumental errors. The analyses were performed under atmospheric pressure, without any preparation. The following parameters were used: 13 kV and 45 μA. All spectra were collected with a fixed measurement time of 60 s. The calibration of this instrument was performed using samples with a phosphorus concentration measured by ICP-AES. Therefore, a correlation curve was established (with a high correlation coefficient, R^2^ = 0.9975) to convert the maximum intensity of the Kα peak into phosphorus mass percentage (Equation (1), Appendix A).
(1)Kα peak intensity=2025.9×phosphorus content (wt%)

The phosphorus content of the various samples was analyzed 3 times.

##### Fluorine Content Measurement

FTIR analysis revealed that flax fabrics treated with fluorinated and phosphonated monomers have a common band for the carbonyl groups C=O at 1735 cm^−1^ (Figure 3). Drying the samples at 60 °C for 24 h was performed to remove the absorbed water and to properly use the -OH band as a reference to compare the spectra of the different samples. In fact, this band was used as a reference because it was not present in the spectrum of the polymers, which were chosen for the grafting. The intensity ratio of the two bands noted *I_C_*_=*O*_/*I_OH_* was used to quantify the grafted phosphorus and fluorine contents.

According to Appendix A, the measurement of the fluorine contents requires several steps. The first one involves a calculation of the phosphorus content according to Equation (1) (Appendix A). Then, samples treated only with MAPC1 having a known phosphorus content (determined by ICP-AES) were analyzed by FTIR to determine the intensity *I_C=O_/I_OH_* ratio and plot the calibration curve (Equation (2), shown in Appendix A).
(2)Phosphorus content (wt%)=5.63×IC=O/IOH

The partial intensity *I_C_*_=*O*_/*I_OH_* ratio, noted R1, which corresponds to the phosphonated units grafted from the fabrics treated with fluorinated and phosphonated monomers, was calculated according to Equation (2). The second step consists in assessing by FTIR the samples treated with both monomers to determine their intensity *I_C_*_=*O*_/*I_OH_* ratio (noted R2). This ratio represents the full ratio for flax grafted with fluorinated and phosphonated polymer chains. The difference of both ratios, R2 − R1, makes it possible to calculate the *I_C_*_=*O*_/*I_OH_* ratio due solely to the fluorinated monomer units grafted onto the flax fabrics. A series of samples treated only with M8 were analyzed by calcination followed by ion chromatography to determine their fluorine content and to establish by comparison with the results of FTIR analyses, a calibration curve as illustrated in Appendix A [36]. The grafted fluorine content was calculated according to Equation (3) (Appendix A).
(3)Fluorine content (wt%)=4.15×IC=O/IOH

Subsequently, Equations (4) and (5) enabled assessing the values of concentrations of grafted monomer units from the fluorine and phosphorus contents, respectively (Table 1).
(4)$$\begin{array}{c} Fluorinated\;monomer\;units\\ \;concentration\left(mol/g\right)\\ \;=\frac{Fluorine\;content\left(wt\%\right)}{Monomer\;fluorine\;content\;\left(wt\%\right)\;\times \;Monomer\;molar\;mass\;(g/mol)} \end{array}$$
(5)$$\begin{array}{c} MAPC1\;units\\ \;concentration\left(mol/g\right)\\ \;=\frac{Phosphorus\;content\;\left(wt\%\right)}{MAPC1\;phosphorus\;content\;\left(wt\%\right)\;\times \;MAPC1\;molarmass\;(g/mol)} \end{array}$$

#### 2.3.4. Pyrolysis Combustion Flow Calorimetry (PCFC)

A pyrolysis combustion flow calorimeter (Fire Testing Technology Ltd., East Grinstead, UK) was used to evaluate the fire behavior of treated fabrics at microscale. Samples (2–4 mg) were pyrolyzed at a heating rate of 1 °C/s under nitrogen (100 mL/min) from 80 to 750 °C (anaerobic pyrolysis—Method A according to the standard ASTM D7309). After the pyrolysis, gases were fully oxidized in the presence of a N_2_/O_2_ (80/20) mixture. The heat rate release (HRR) was calculated according to Huggett’s relation (1 kg of consumed oxygen corresponds to 13.1 MJ of released energy) [37]. Each test was performed twice to ensure the reproducibility of the analysis. The peak of heat rate release (pHRR), the temperature at pHRR (Tmax), the total heat release (THR) and the char content were determined.

#### 2.3.5. Cone Calorimetry

The cone calorimeter is a technique to assess the fire behavior of materials at bench scale. These experiments were performed to evaluate the impact of phosphorus content at a heat flux of 35 kW/m^2^. The distance between the radiant cone and the sample was 25 mm. The 10 × 10 cm^2^ fabrics were placed horizontally on a sample holder and were wrapped in aluminum foil. The bottom surface was insulated with rock wool. A metal grid having a mesh size of 1.8 × 1.8 cm^2^ and a thickness of 0.2 cm was placed on the upper surface of the sample to prevent deformation of the fabric during the test. Air flow was fixed at 24 L/s. The samples decomposed and released combustible gases, which ignited in the presence of a spark. The heat release rate (HRR) was also calculated according to Huggett’s relation [37]. The peak heat release rate (pHRR), time to ignition (TTI), total heat released (THR) and final residue content were determined.

#### 2.3.6. Contact Angle Measurements

A KRÜSS-type goniometer (opsira, Nürnberg, Germany)was used to measure the contact angle of liquid drops formed on the surface of the flax fabric samples. For the hydrophobicity assessment, water was used as the contact angle measuring liquid (WCA). For the oleophobicity, diiodomethane was used to lead to DCA. After adjustment of the deposition level, a drop of 9 μL of water or 1.5 μL of diiodomethane was placed on the surface of the treated fabrics. The baseline used to measure the contact angle was determined for each analysis by the KRÜSS ADVANCE software version 4.0. For each sample, five measurements were performed to ensure reproducibility.

#### 2.3.7. Sliding Angle Measurements

Measurements of sliding angles of hydrophobic fabrics were carried out using a set up realized in our laboratory. The sample was placed on a flat substrate and then a drop of deionized water of 30 µL was put onto the modified grafted fabrics. The substrate was then progressively inclined at angles ranging between 0 and 90°. The sliding angle was determined as the angle value for which the water drop slides off the fabric surface. For each sample, four measurements were performed.

## 3. Results and Discussion

This study deals with the development of multifunctional fabrics endowed with flame retardancy and hydro- and oleophobicity properties. It concerns a one-step procedure using the pre-irradiation method with two different monomers. MAPC1, which contains a phosphonated function, and chosen as the FR monomer to improve the flame retardancy of flax fabrics. The (meth)acrylic monomers, M4, AC6 and M8, bearing perfluorinated groups of different lengths (4, 6 and 8 carbons) were used for the modification of the surface energy of the fabrics to make them hydro- and oleophobic. The influence of the combined fluorinated and phosphonated monomers on the studied properties is developed in this study.

### 3.1. Grafting of Fluorinated and Phosphorus Polymers onto Irradiated Flax Fabrics

#### 3.1.1. FTIR Analysis

The grafting of polymer chains using M8 and MAPC1 monomers alone or in combination (50/50 wt%) onto irradiated flax fabrics at 100 kGy was examined using infrared spectroscopy (Figure 3). The observed bands at 1735 and 1146 cm^−1^ correspond to the C=O carbonyl and C-O-C ether groups, respectively [38,39]. For M8 and MAPC1 combination as for MAPC1 alone, the FTIR spectra show the presence of two bands at 1250 and 790 cm^−1^ attributed to P=O and P-O-C, respectively [38,40]. Moreover, it was also observed the presence of the characteristic bands of fluorinated polymer chains at 1200 cm^−1^ corresponding to the C-F bonds when the monomers were combined and also for the grafting of M8 alone [39,41,42]. Two bands of medium intensity appeared at 655 and 703 cm^−1^ resulting from a combination of rocking and wagging vibrations of the CF_2_ groups [29,43]. These results highlight the one-step grafting of both phosphorus and fluorinated monomers onto irradiated flax fabrics.

The same results were obtained for the other fluorinated monomers when combined with MAPC1 at different ratios and for doses of 20 and 100 kGy (Table 1).

#### 3.1.2. Reactivity of Monomers Used in Radiation-Induced Grafting Polymerization

Table 1 summarizes the various fluorine (FC) and phosphorus (PC) contents for flax fabrics treated with M4, AC6 or M8 in combination with MAPC1. The fluorinated to phosphonated monomer molar ratio was noted as F/P. The initial (i.e., in the reaction solution) F/P and the final F/P of modified flax fibers were compared. It is noted that the dose and the monomer concentration directly impact the grafted fluorine and phosphorus contents whatever the monomer combination (Table 1). For an initial M8/MAPC1 mixture (50/50 wt%) at a dose of 20 kGy, 3.82 and 0.56 wt% of fluorine and phosphorus contents were achieved, respectively. For a similar monomer combination but at a dose of 100 kGy, these contents were 8.04 and 0.77 wt%, respectively. For a dose of 20 kGy and M8/MAPC1 monomer ratios of 20/80, 50/50 and 80/20, the fluorine content increased from 0.22 to 3.82 and 4.55 wt% while phosphorus content increased from 0.46 to 0.56 but then decreased to 0.29 wt%. Overall, the grafting efficiency of both comonomers seems to increase with the dose of irradiation and their proportion in the reaction solution.

Appendix A indicates that the concentration of grafted fluorinated monomer increased with the increase in the F/P molar ratio in the reaction solution except for a few samples. The concentration increased from 0.16 to 2.99 × 10^−4^ mol/g for 20/80 and 50/50 M8/MAPC1 mixtures, respectively, followed by a slight decrease to 2.05 × 10^−4^ mol/g for the 80/20 mixture. For 100 kGy and an 80/20 mixture, the treatment with M4/MAPC1 and AC6/MAPC1 resulted in identical concentrations of fluorinated monomer units of about 0.25 × 10^−4^ mol/g. On the other hand, at a low dose of 20 kGy and for 20/80 and 50/50 mixtures, low molar concentrations of the fluorinated monomer were obtained for AC6/MAPC1 (0.06 and 0.07 × 10^−4^ mol/g) in comparison with M4/MAPC1 (0.15 and 0.16 × 10^−4^ mol/g). These results are not in agreement with previous work in homopolymerization conditions, where the AC6 monomer was more efficiently grafted than M4 [36], as also observed by Guyot et al. [44]. This is probably due to a disruption of the AC6 reactivity in the presence of MAPC1. M8 grafting was probably less affected by MAPC1 presence and showed a significant grafting efficiency compared to those of M4 and AC6. Indeed, the grafting values obtained in the presence of MAPC1 are close to those of M8 alone [36]. However, it was also noted that the concentration of grafted phosphonated monomer decreases with the increase in the F/P molar ratio in the impregnation solution (Appendix A), except in the case of grafting of the M8/MAPC1 mixture, where the grafted phosphonated concentration first increased from 1.48 to 1.81 × 10^−4^ mol/g at 20 kGy and from 1 to 2.48 × 10^−4^ mol/g at a dose of 100 kGy, when the initial F/P molar ratio increased from 0.10 to 0.39, respectively. Then, a decrease was observed for the initial F/P molar ratio of 1.56, with values of 0.94 and 0.77 × 10^−4^ mol/g achieved for the 20 kGy and 100 kGy doses, respectively. These results show a higher grafting efficiency for M8 than for MAPC1. Appendix A represents the final F/P monomers molar ratio in the grafted flax vs. the initial one in the reaction solution. The results indicate that the final F/P-monomer ratio is lower than the initial one for the M4/MAPC1 and AC6/MAPC1 mixtures, which is more visible for the highest values of the initial ratio. This can be assumed to arise from a higher change of M4 and AC6 polymerization behaviors in the presence of MAPC1. On the other hand, M8/MAPC1 mixture revealed a distinct behavior, where the final F/P-monomer ratio in the treated fabrics is higher than the initial one, except for flax irradiated at 20 kGy with an initial ratio of 20/80 wt%. To conclude, the grafting efficiency of the fluorinated monomers in the presence of MAPC1 seems to depend on their structure, especially on the fluoroalkyl length. The efficiency for the grafting of the fluorinated monomer is classified in the following increasing order: AC6~M4 < M8. MAPC1 therefore displays a significant reactivity compared to M4 and AC6.

#### 3.1.3. Localization of the Fluorine and Phosphorus Elements in the Modified Flax Fibers

The longitudinal and cross-sections of flax fibers irradiated at 100 kGy and treated with different combinations of MAPC1 and/or fluorinated monomers were analyzed by SEM-EDX. This technique enabled us to study the evolution of the flax fiber morphology with the treatment and to evaluate the distribution of the phosphorus and fluorine elements within their section.

The SEM pictures were performed to investigate and to compare the morphology of treated and untreated flax fabrics (Figure 4). A smooth texture is noted for pristine flax fibers (Figure 4a). For fabrics irradiated at 100 kGy and treated with the M4/MAPC1 and AC6/MAPC1 (50/50) mixtures, a homogeneous polymer coating on the elementary fiber surface (Figure 4b and c, respectively) was formed. In the case of M8/MAPC1 (50/50), the formation of a rough polymer coating composed of polymer spheres partially fused together on the surface of the flax elementary fibers is observed (Figure 4d,e).

The SEM-EDX mapping of these fibers exhibited that the spheres at the fiber surface contain the fluorine element only (Figure 4e1). These structures based on fluorinated polymer at the surface of elementary fibers are assumed to be from a dispersion polymerization of M8 in methanol during the grafting reaction [45,46]. Similar results were observed in a previous study on the radiografting of M8 onto flax fabrics by a pre-irradiation procedure [36].

The SEM-EDX analyses of the cross-section for the treated fabrics are presented in Figure 5. For fabrics irradiated at 100 kGy and treated only with MAPC1 for 24 h, it is observed that the phosphorus atoms are homogeneously located on both the surface and the bulk of the elementary fibers (Figure 5c). This phosphorus distribution is in good agreement with the results obtained in a previous study, which used a similar procedure involving water as the solvent for the grafting reaction [27]. On the contrary, under the same conditions as those of MAPC1, for the grafting of M8 alone, the fluorine element is only present at the surface of the elementary fibers (Figure 5e).

The one-step grafting of both fluorinated and phosphorus monomers resulted in a different distribution of the chemical elements in the cross-section of flax fibers. The combined grafting of M8 and MAPC1 (50/50) with fabrics irradiated at 100 kGy revealed that fluorine element is present only at the surface of the elementary flax fibers in a high concentration (FC = 8.04 wt%) (Figure 5h), while phosphorus was present in both the bulk and on the surface (PC = 0.77 wt%) (Figure 5i). The localization of these elements is the same as that observed for the separate grafting reactions of these two kinds of monomers. The limited diffusion of the M8 monomer into the elementary fibers is confirmed even when this monomer is combined with MAPC1.

Unlike the M8/MAPC1 mixture, the distributions of fluorine and phosphorus elements after treatment of fibers irradiated at 100 kGy with AC6/MAPC1 (FC = 0.72, PC = 0.17 wt%) and M4/MAPC1 (FC = 0.33 wt%, PC = 0.26 wt%) combinations are identical. Fluorine (Figure 5k,n, respectively) and phosphorus (Figure 5l,o, respectively) elements are homogeneously located in the bulk and on the surface of the elementary fibers.

### 3.2. Hydro- and Oleophobic Properties and Water Repellency of Treated Flax Fabrics

After modification with the different mixtures of F/P monomers, the hydro- and oleophobic properties and the water repellency of the treated flax fabrics were investigated. Table 2 summarizes the water (WCAs) and diiodomethane (DCAs) contact angles and the sliding angles (SAs) for the treated fabrics.

The hydrophobic and oleophobic properties of the fabrics treated with the M4/MAPC1, AC6/MAPC1 and M8/MAPC1 mixtures for different ratio values were evaluated by measuring the WCA and DCA, respectively. The results were correlated to the FC in the modified flax fibers (Figure 6). Whatever the proportion of grafted MAPC1, it can be observed for these different samples that the WCA (Figure 6a) and DCA (Figure 6b) values increase significantly with FC from 0.16 to 0.30 wt%. Above FC of ca. 0.30 wt% until the maximum FC of 8.04 wt% (M8/MAPC1 (50/50)—100kGy), the WCA and DCA values remain stable at ca. 150° and 130°, respectively. In a previous study concerning the grafting of fluorinated monomers onto flax fibers by the pre-irradiation method [36], the same results were observed with WCA and DCA data, which reached a maximum value and remained stable from a fluorine rate greater than or equal to 0.3 wt%. These high values of WCA and DCA seem to prove that even in the presence of MAPC1 units, hydrophobicity and oleophobicity can be achieved for the modified fibers. These surface properties are controlled by the fluorine concentration but also by the F/P-monomer ratio in the treated fabric. Indeed, when this ratio is too low, the sample remains hydrophilic and oleophilic. Irradiation at a dose of 20 kGy and treatment with a low fluorinated monomer concentration (M4/MAPC1, AC6/MAPC1 or M8/MAPC1 = 20/80) resulted in hydrophilic and oleophilic behavior of flax fabrics due to a too low initial F/P ratio. Indeed, similar FCs of 0.26 and 0.27 wt% obtained for fabrics irradiated at 20 kGy and treated with M4/MAPC1 (20/80) and (50/50), respectively, led to different tendencies. An oleophilic (0°) behavior was observed in the first case with a high phosphorus content of 0.45 wt%, while the second one led to hydrophobic (124°) and oleophilic (74°) tendencies with a low phosphorus content of 0.10 wt%. This evidences that the phosphorus element may affect the input of the fluorinated monomer and thus may change the surface properties of such modified fabrics.

For other fluorinated and phosphorus monomer ratios and irradiation doses, all samples produced were hydrophobic and oleophobic except the fabric irradiated at 20 kGy and treated with M4/MAPC1 50:50 wt%. In this case, the modified sample showed a low DCA (74°). At an identical (50/50) monomer ratio and a dose of 20 kGy, FC values of 0.27, 0.18 and 3.82 wt% were obtained for M4, AC6 and M8, respectively. The resulting WCA/DCA values were 124°/74°, 129°/121° and 146°/134°, respectively. At a higher dose of 100 kGy, greater FC values of 0.33, 0.72 and 8.04 wt%, were obtained with WCA/DCA values of 140°/129°, 135°/125° and 151°/131°, respectively. This reveals that the best surface properties were obtained with an M8/MAPC1 grafting, as evidenced by superhydrophobic fabrics also produced with the M8/MAPC1 (50/50) treatment as with the treatment with only M8 (WCA = 150° and DCA = 136°) [36].

For fabrics irradiated at 20 kGy and treated with M4/MAPC1, AC6/MAPC1 and M8/MAPC1 with an 80/20 ratio, FC values of 0.24, 0.37 and 4.55 wt% of FCs were obtained. WCA/DCA of 133°/125°, 145°/130° and 147°/131° were achieved, respectively. For the same monomer ratio at 100 kGy, a slight increase in FC was noted from 0.24 to 0.40 wt% for M4/MAPC1 (WCA = 142°, DCA = 130°), from 0.37 to 0.63 wt% for AC6/MAPC1 (WCA = 142°, DCA = 131°) and from 4.55 to 6.63 wt% for M8/MAPC1 (WCA = 149°, DCA = 132°). At a high initial F/P ratio, the water and diiodomethane contact angles reached stability even by increasing the FC.

The water repellency of the treated fabrics was also studied by measuring the sliding angles (SAs) (Table 2). Figure 7 represents the evolution of the SAs versus the FC for the treated flax fabrics. The SA values decrease rapidly from 90° to 10° with the increase in the FC from 0.2 to 2.0 wt%, followed by a plateau for values higher than 2 wt% until 10 wt%. For M4/MAPC1 mixtures, the SA ranged from 90 to 40°, while for AC6/MAPC1, the smaller sliding angle achieved was 30°. The same SAs window was reached in the case of grafting of fluorinated monomers M4 and AC6 alone [36]. For fabrics treated with M8/MAPC1 mixtures, small SA values lower than 13° (and 10° for grafting M8 alone) were obtained corresponding to satisfactory water-repellency properties [32,47]. These results show that the SA values are directly impacted by the FC for treatments with the three fluorinated monomers combined with MAPC1.

### 3.3. Flame-Retardant Properties of Treated Flax Fabrics

The introduction of the phosphorus element onto the grafted flax should improve its fire resistance [23,24,48]. Flame retardancy of the treated flax fabrics was assessed by pyrolysis combustion flow calorimetry (PCFC), and the main data are presented in Table 3.

The heat release rate (HRR) curves for the different samples, untreated and treated, are gathered in Figure 8 and Appendix A (ESI). For pristine flax fabric, the peak of heat release rate (pHRR) occurs at about 370 °C with a value of about 230 W/g and total heat release (THR) close to 9 kJ/g. These results are in good agreement with previous works [6,7]. Furthermore, grafting of M4, AC6 or M8 onto flax fabrics irradiated at 20 and 100 kGy revealed no noticeable modification of their fire behavior (Appendix A).

The grafting with only MAPC1 polymer chains onto flax fabrics irradiated at 20 kGy (Figure 8a) resulted in a 0.22 wt% of phosphorus content and produced decreases in both pHRR (from 230 to 131 W/g) and pHRR temperature (from 370 to 312 °C). The THR value also decreased to 6.8 kJ/g while the char residue increased to 21 wt% versus 11 wt% for pristine flax fabrics. At 100 kGy (Figure 8b), higher phosphorus content was achieved (1.77 wt%), and sharp decreases in pHRR (from 230 to 47 W/g), pHRR temperature (from 370 to 255 °C) and THR (from 9 to 2.9 kJ/g) were noted. Char content increased to 40 wt%.

Flame retardancy properties were also assessed for the samples treated with fluorinated and phosphonated monomer mixtures at various ratios. The fire behavior of fabrics treated with the M4/MAPC1 mixture is shown in Figure 8. This mixture was selected for toxicity reasons while M4 contains a short fluoroalkyl chain. For flax irradiated at 20 kGy and treated with the M4/MAPC1 80/20 mixture, 0.24 and 0.07 wt% of fluorine and phosphorus contents were achieved, respectively. Grafting of 0.07 wt% of phosphorus led to a slight decrease in values of pHRR and temperature of pHRR (208 W/g and 329 °C, respectively). However, the THR remained the same as that of the pristine fabrics with a similar char content of about 13 wt%. This weak evolution is due to the low phosphorus amount grafted onto the flax fibers (Figure 8a). Under the same grafting conditions but with a 100 kGy irradiation dose (Figure 8b), higher fluorine and phosphorus contents of 0.40 and 0.28 wt% were reached, respectively. Values of pHRR, temperature of pHRR, THR and char content of 122 W/g, 302 °C, 6.9 kJ/g and 18 wt% were obtained, respectively. When the MAPC1 amount increased in the reaction solution, as for the M4/MAPC1 50/50 mixture, 0.33 and 0.26 wt% of FC and PC were reached, respectively. This sample displayed a pHRR value of about 106 W/g at 304 °C, a THR close to 5.4 kJ/g and a residue rate of 24 wt%. For a monomer ratio with PC higher than the fluorine one (M4/MAPC1 = 20/80), quasi-identical fluorine and phosphorus contents were obtained (0.26 and 0.23 wt%, respectively) for flax fabrics and resulted in a pHRR of 121 W/g at a temperature of 312 °C. The measured THR was ca. 5.6 kJ/g while char content was 22 wt%. The same evolutions were noted for the AC6/MAPC1 (Appendix A, Appendix A) and M8/MAPC1 mixtures (Appendix A).

Main PCFC data are plotted vs. the phosphorus content in Figure 9. The intensity of pHRR decreased systematically from 230 W/g to 47 W/g (Figure 9a) when phosphorus content increased. Because of the early decomposition of cellulose, the pHRR temperature decreased from 370 °C to 255 °C (Figure 9b). In addition, THR decreased from 9.0 kJ/g to 2.8 kJ/g, due to the partial trapping of carbon into the condensed phase (Figure 9c). Indeed, the char content increased from 11 wt% to ca. 40 wt% (Figure 9d). These results are attributed to the fact that the phosphonated group in MAPC1 units acts as a flame retardant. With the temperature increase, this group decomposes causing the formation of phosphoric acid, which can induce a phosphorylation of the primary hydroxyl group of cellulose to form a phosphorus ester [23,49]. These esters catalyze the dehydration of cellulose at low temperature, leading to char formation [49]. Therefore, charring is assisted by the presence of phosphorus, leading to higher residue yield and lower THR but decreased thermal stability compared to pristine flax fabrics. The three fluoro-phosphonated mixtures produced identical results and the flammability at the microscale is mainly impacted by the phosphorus content (Appendix A, ESI). The results indicate the same tendency as reported by Hajj et al. [7] for simultaneous radiografting procedures and also in our own work on pre-irradiation polymerization of MAPC1 alone in water [27]. In other words, the flame retardancy at microscale depends only on phosphorus content and is not affected by the presence of fluorinated groups. The comparison with these results also proves that the grafting of MAPC1 in water made it possible to reach higher phosphorus content than with methanol.

From the different results obtained a superhydrophobic fabric was obtained from M8 only and from high irradiation dose (100 kGy) and monomer concentration. However, due to the toxicity, bioaccumulation, persistency and mobility of longer fluorinated alkyl groups (C8 and C6) [17,20,30,31,32,33,34,35], M4 was preferred for further study. The treatment with M4/MAPC1 (50/50 mixture) and flax irradiated at 100 kGy was chosen as the suitable conditions to produce a multifunctional fabric combining hydrophobic, oleophobic and flame-retardant properties.

Flax fabric treated under the appropriate conditions has been prepared again, in larger quantities, with 0.49 wt% and 0.77 wt% of fluorine and phosphorus contents, respectively. The modified flax fabric and the pristine fabric were then analyzed with a cone calorimeter apparatus to evaluate the effect of the grafting. The main flammability data of these samples are listed in Table 4.

Compared to pristine flax, the fabrics modified with the M4/MAPC1 50/50 mixture induce a significant decrease in ignition time (TTI) from 28 s to 14 s (Figure 10). No evolution of pHRR was observed for the treated fabric in comparison with that of the pristine fabric (98 and 102 KW/m^2^, respectively). Actually, the pHRR for thermally thin materials as fabrics was mainly dependent on the sample mass and the heat of combustion. Indeed, in another work [50], a phenomenological model to calculate the pHRR of thermally thin materials was proposed. Using this model and considering the data listed in Table 4, pHRR was found to be 122 and 91 kW/m^2^ for untreated and treated fabrics, respectively. This is in acceptable agreement with the experimental values.

The total heat release (THR) also decreased significantly from 15.7 kJ/g to 10.7 kJ/g. The final residue resulting from this test is displayed in Figure 11. It was noted that for pristine flax fabric (absence of phosphorus) no residue was obtained while in the case of treated fabrics, a significant residue rate of 17 wt% was produced. These results are in good agreement with the work of Hajj et al. [7] on the radiografting of vinyl phosphonic acid (VPA) onto flax fibers by the simultaneous method. For a phosphorus content of 1.1 wt% and a heat flux of 35 kW/m^2^, TTI decreased from 27 s to 12 s, the pHRR decreased from 100 to 80 kW/m^2^, while the residue increased from 7.0 to 31.5 wt%. In our previous work, similar results were observed [27]. Fabrics irradiated at 10 and 100 kGy and modified with 10 wt% MAPC1 in water for 24 h at 80 °C were prepared and phosphorus contents of 1.4 and 2.4 wt% were reached, respectively. At a heat flux of 35 kW/m^2^, the ignition time decreased from 27 s for untreated flax fabrics to 14 and 16 s for fabrics irradiated at 10 and 100 kGy, respectively. The pHRR decreased from 91 kW/m^2^ to 72 kW/m^2^ and 78 kW/m^2^. THR also decreased after treatment from 11.3 kJ/g to 8.9 and 7.9 kJ/g. The final residue resulting after the test was ca. 19.1 and 25.5 wt% for 1.4 and 2.4 wt% of phosphorus, respectively. These results evidence that the flame-retardant properties of the treated fabrics are mainly controlled by the presence of grafted phosphorus.

## 4. Conclusions

In this work, multifunctionalized flax fabrics combining flame-retardant, hydrophobic and oleophobic properties were prepared in a one-step radiation-induced copolymerization. Indeed, the use of a combination of a phosphorus-containing methacrylic monomer with (meth)acrylic monomers bearing different perfluorinated lengths (M4, AC6 or M8) made the grafting of polymer chains possible with appropriate properties. The successful multigrafting of flax fabrics was confirmed by both FTIR and SEM-EDX measurements. The resulting fabrics presented simultaneously flame-retardant, hydrophobic and oleophobic properties depending on the grafting rate of fluorinated and phosphonated monomers. The SEM images showed the formation of a smooth polymer coating in the case of M4/MAPC1 and AC6/MAPC1 mixtures. However, for the treatment with the M8/MAPC1 mixture, a rough polymer layer appearing as spherical particles partially fused together at the fiber surface was observed. SEM-EDX mapping revealed that phosphorus and fluorine atoms were homogeneously distributed in the bulk and on the surface of the elementary flax fibers for treatments with M4/MAPC1 and AC6/MAPC1 mixtures. However, when the M8/MAPC1 mixture was used, phosphorus was located in the bulk and on the surface of the elementary fibers, while the fluorine element was present only on the surface. This difference in selectivity was assumed to be due to the length of the perfluorinated group of the fluoro monomer, which changes its affinity with the reaction solvent and with the different parts of the flax fibers. The pre-irradiation procedure with the M4/MAPC1, AC6/MAPC1 or M8/MAPC1 mixtures produced multifunctional fabrics that were flame retardant, hydrophobic and oleophobic in most cases. However, for a low irradiation dose (20 kGy) and a low fluorinated monomer concentration, the modified fabrics remained hydrophilic and oleophilic. Fabrics irradiated and treated with M4 in combination with MAPC1 showed promising results. Indeed, for flax irradiated at 100 kGy and treated with a 50/50 mixture, values of FC (0.33 wt%) and PC (0.26 wt%) were obtained, as well as high WCA (149°) and DCA (128°). It was evidenced that the hydrophobicity and oleophobicity of modified fabrics were managed by the final fluorine content and the ratio between the grafted fluorinated and phosphonated monomers. Similarly, as observed in this study and in previous works, the flame retardancy of functionalized flax fabrics was controlled primarily by the phosphorus content. It seems that for the different combinations, the simultaneous presence of both two monomers in the modified flax weakly affects the respective function of each.

Further to this work, the impact of the affinity of the fluorinated monomers with the reaction solvent should be better evaluated in order to control the localization of the grafting or the texturing of the polymer coating formed on the fibers. A study of the mechanical properties of flax fabrics that have been functionalized would also allow evaluating if the grafting induces a reinforcement or embrittlement of the fibers. It would also be particularly interesting to study the washing resistance of the treatments. Indeed, the treatment developed allows grafting covalently the phosphorus and fluorinated monomers, and it will thus be necessary to evaluate its resistance to washing in time.

## Figures and Tables

**Figure 1 polymers-15-02169-f001:**
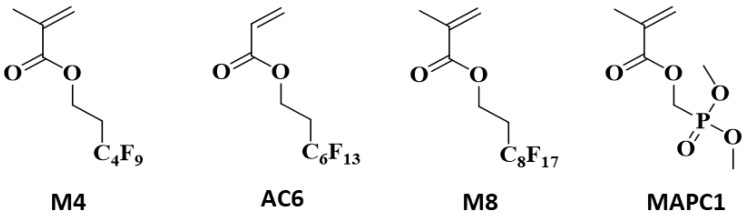
Chemical structures of the fluorinated (M4, AC6 and M8) and phosphonated (MAPC1) monomers used in this study.

**Figure 2 polymers-15-02169-f002:**
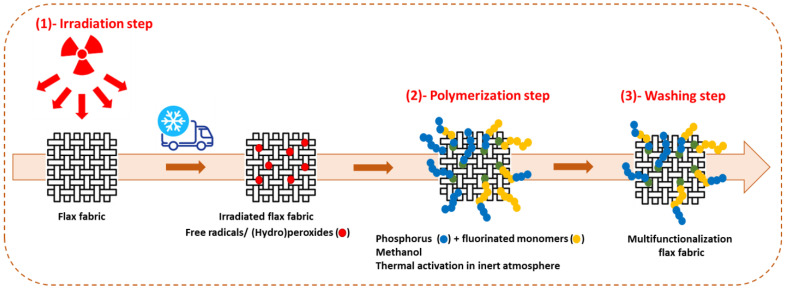
Scheme of the general procedure used for pr-irradiation polymerization.

**Figure 3 polymers-15-02169-f003:**
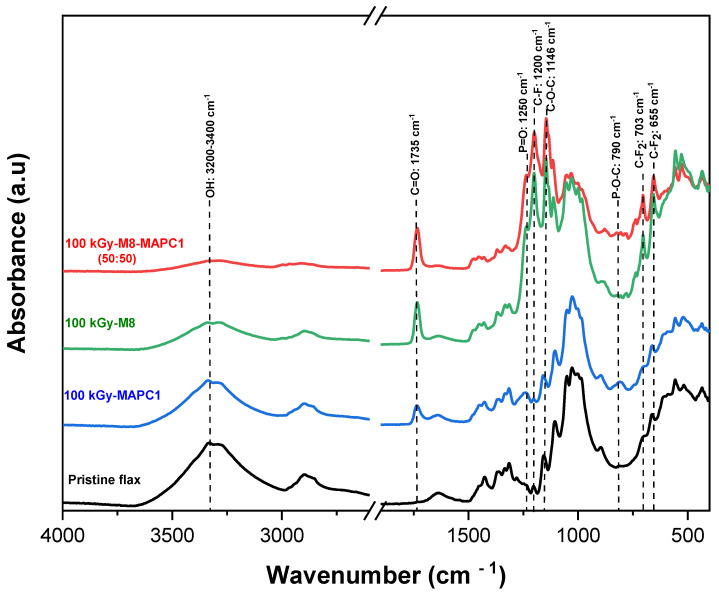
FTIR spectra of pristine flax fabric, irradiated flax fabrics at 100 kGy and treated with MAPC1, M8 and with M8/MAPC1 (50:50) mixture (10 wt%_65 °C_24 h).

**Figure 4 polymers-15-02169-f004:**
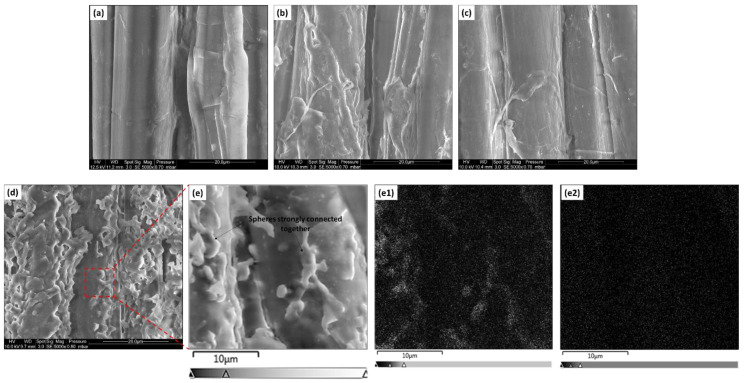
SEM images of (**a**) pristine flax fibers and irradiated flax fibers at 100 kGy and treated with (**b**) M4/MAPC1, (**c**) AC6/MAPC1 and (**d**,**e**) M8/MAPC1 (50/50) mixtures. Magnification ×5000 for (**a**–**d**) and ×10,000 for (**e**). (**e1**,**e2**) are the corresponding EDX fluorine and phosphorus mappings, respectively, for fibers treated with M8/MAPC1 (50/50) mixture.

**Figure 5 polymers-15-02169-f005:**
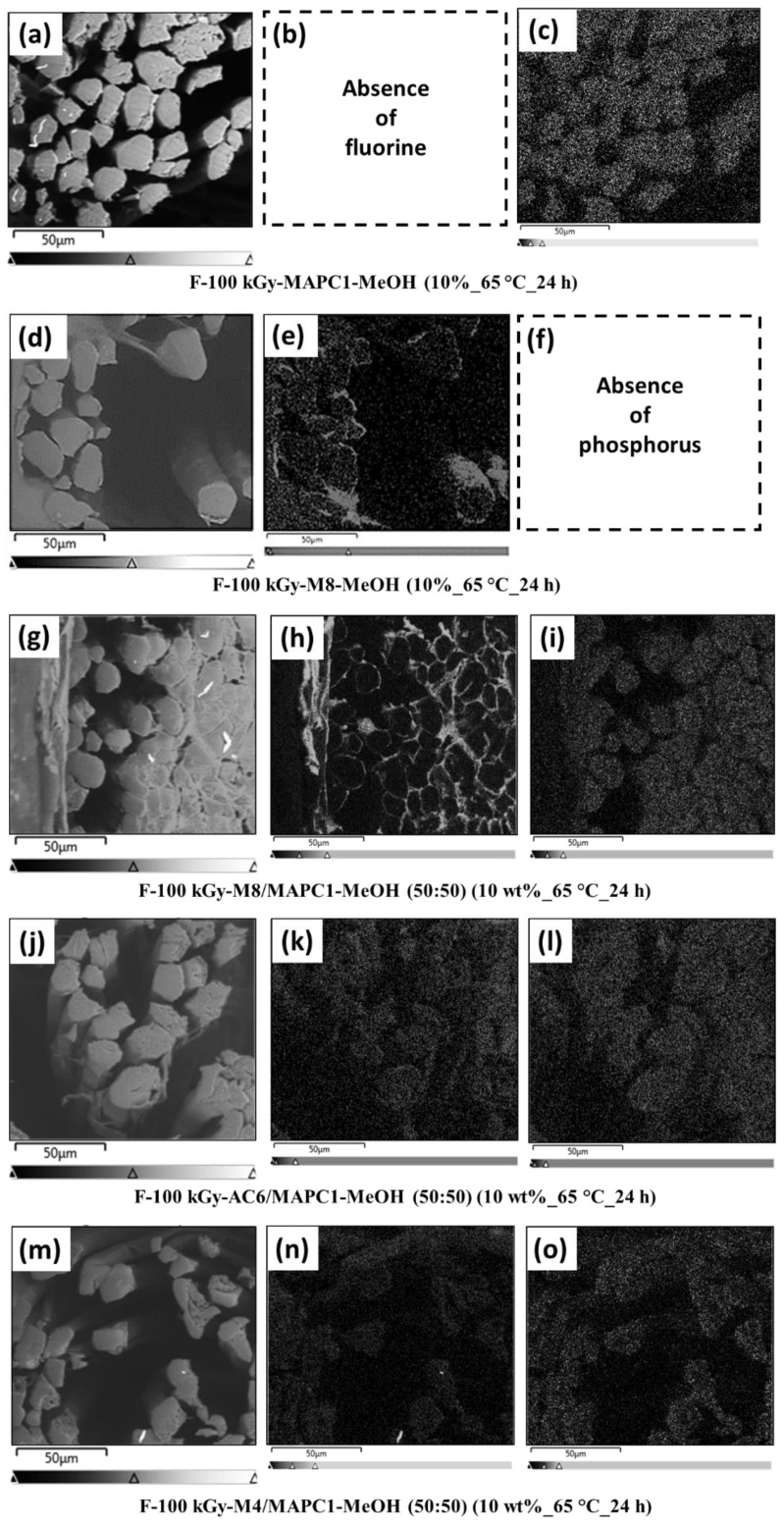
SEM images of irradiated flax fabrics at 100 kGy and treated with (**a**) MAPC1, (**d**) M8, (**g**) M8/MAPC1, (**j**) AC6/MAPC1, (**m**) M4/MAPC1 (50/50 wt% mixtures), (**b**,**e**,**h**,**k**,**n**) are the corresponding EDX fluorine mapping, (**c**,**f**,**I**,**l**,**o**) are the corresponding EDX phosphorus mapping.

**Figure 6 polymers-15-02169-f006:**
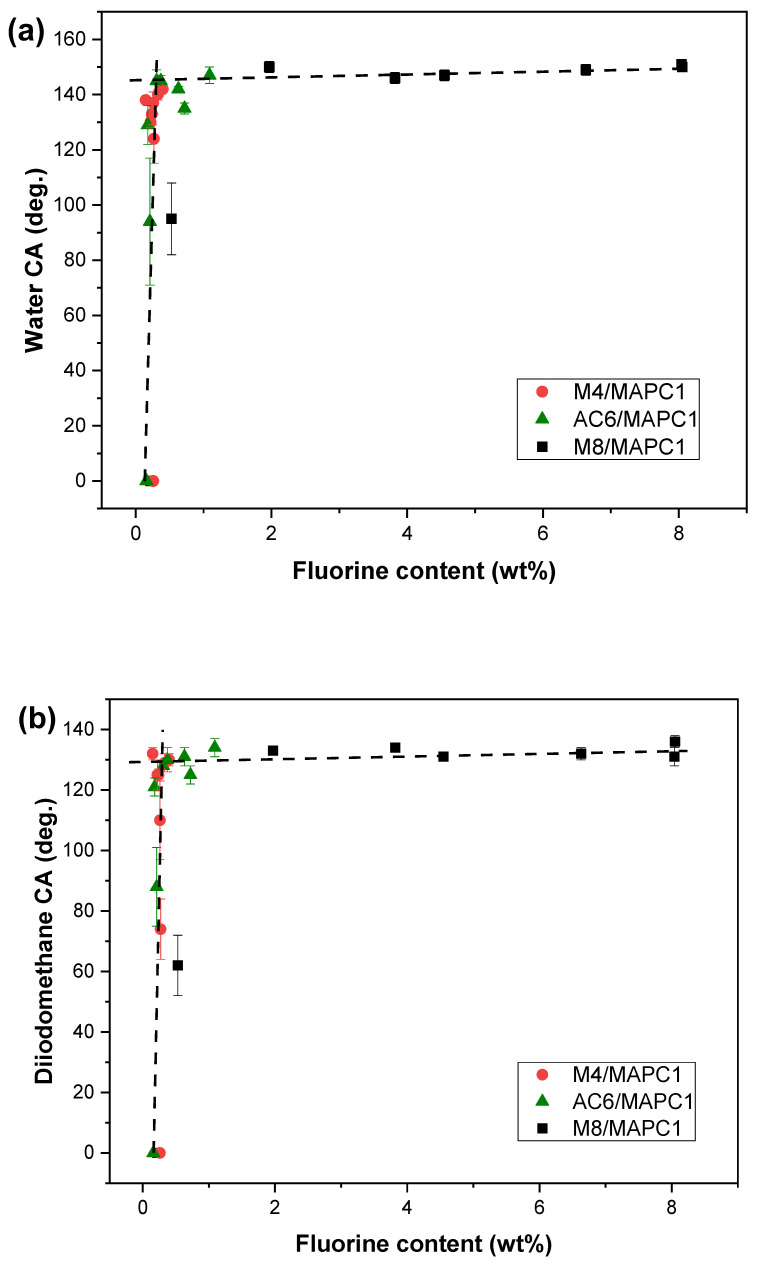
Evolutions of (**a**) water and (**b**) diiodomethane contact angles versus grafted fluorine content for various flax grafted by M4/MAPC1 (●), AC6/MAPC1 (▲) and M8/MAPC1 (◼) (dotted lines are guidelines for eyes).

**Figure 7 polymers-15-02169-f007:**
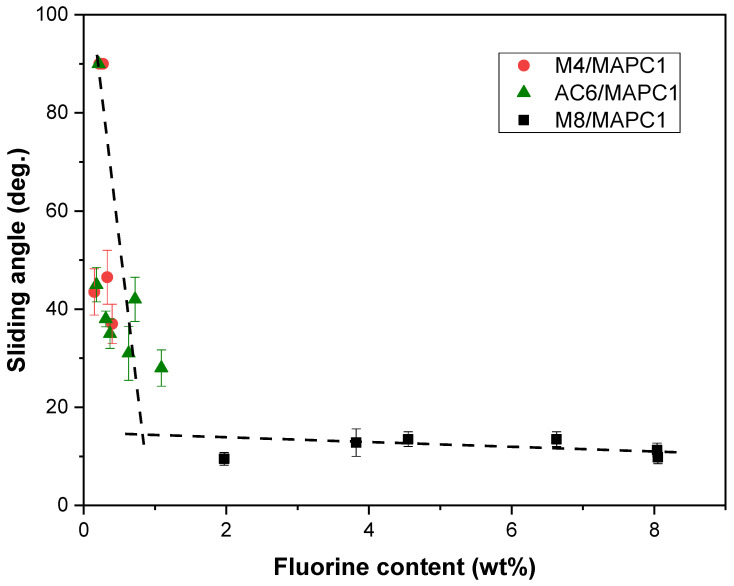
Evolutions of the sliding angles according to the fluorine content for various flax treated with M4/MAPC1 (●), AC6/MAPC1 (▲) and M8/MAPC1 mixtures (◼) (dotted lines are guidelines for eyes).

**Figure 8 polymers-15-02169-f008:**
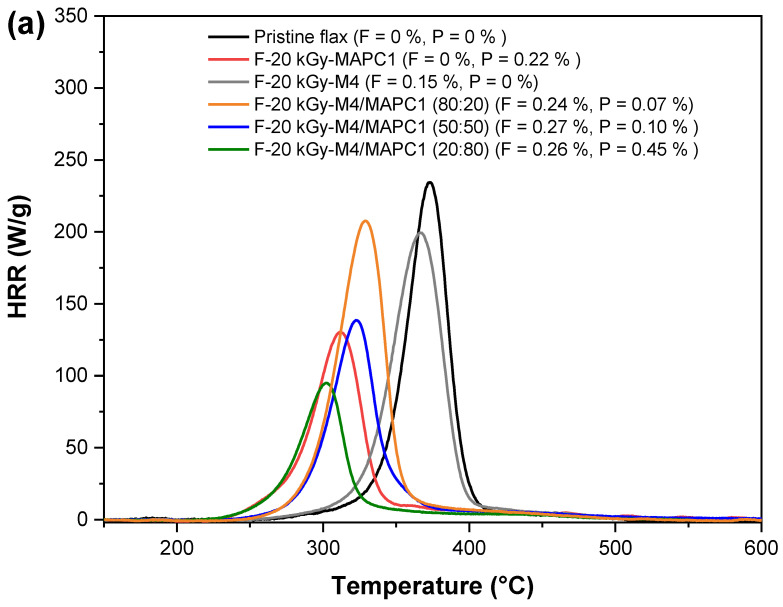
Pyrolysis combustion flow calorimetry (PCFC)-HRR curves (anaerobic pyrolysis) of flax fabrics irradiated at (**a**) 20 kGy and at (**b**) 100 kGy and treated with M4/MAPC1 at various initial monomer ratios.

**Figure 9 polymers-15-02169-f009:**
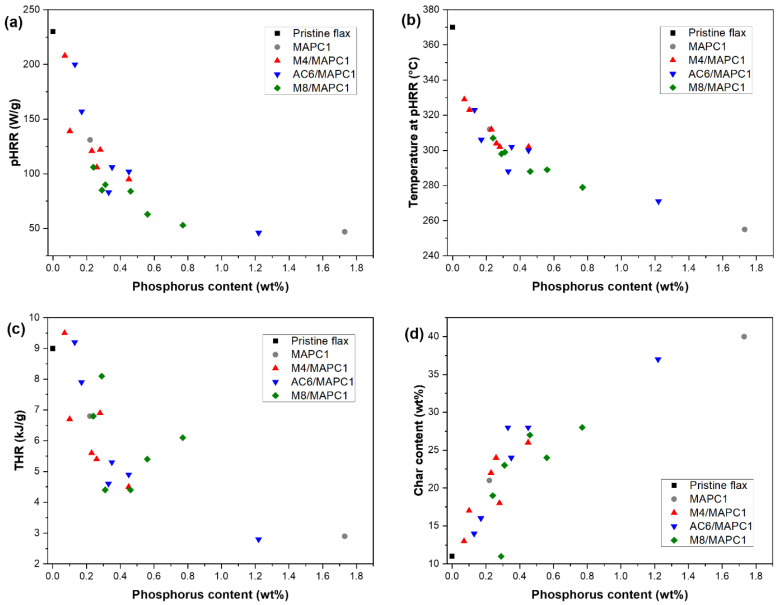
Effect of phosphorus content in the treated flax fabrics on (**a**) pHRR, (**b**) temperature at pHRR, (**c**) THR and (**d**) char content from PCFC analysis.

**Figure 10 polymers-15-02169-f010:**
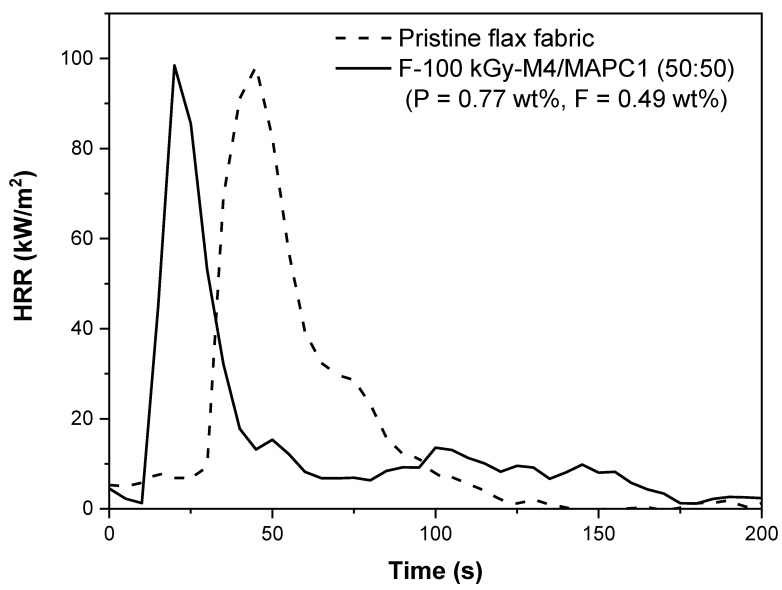
HRR curves at heat flux of 35 kW/m^2^ for pristine flax fabric (dotted line) and for irradiated flax at 100 kGy and modified with an M4/MAPC1 50:50 mixture (10 wt%_65 °C_ 24 h) (full line).

**Figure 11 polymers-15-02169-f011:**
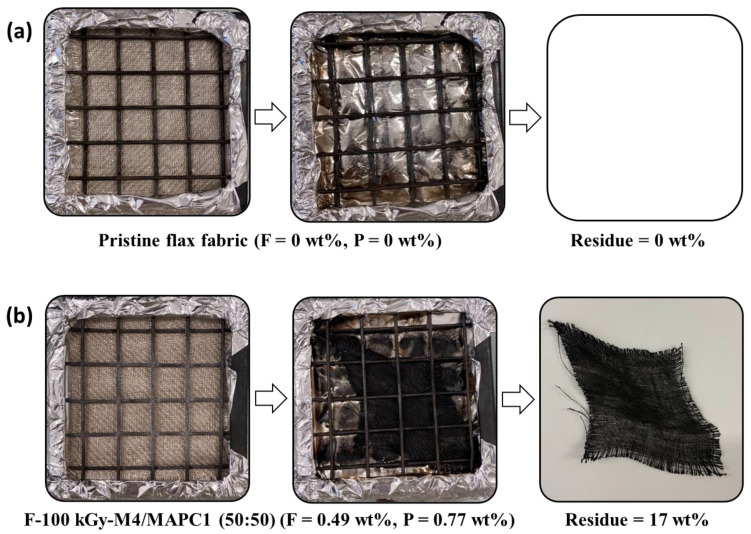
Pictures of residues after cone calorimeter tests of (**a**) pristine flax and (**b**) irradiated flax at 100 kGy and treated with M4/MAPC1 50/50 mixture for 24 h.

**Table 1 polymers-15-02169-t001:** Fluorine and phosphorus contents for all treated flax fabrics and molar F/P ratio (initial and final).

Monomers Combination	Monomers Ratio (F/P)	Dose (kGy)	F-Content (wt%) ^a^	P-Content (wt%) ^b^	F-Monomer Concentration (10^−4^ mol/g) ^c^	P-Monomer Concentration (10^−4^ mol/g) ^d^	Initial F/P Monomers Molar Ratio	Final F/P Monomers Molar Ratio
MAPC1	0:100	20	-	0.22	-	0.71	-	-
	0:100	100	-	1.73	-	5.58	-	-
M4	100:0	20	0.15	-	0.09	-	-	-
	100:0	100	0.22	-	0.13	-	-	-
AC6	100:0	20	0.31	-	0.13	-	-	-
	100:0	100	1.09	-	0.44	-	-	-
M8	100:0	20	1.97	-	0.61	-	-	-
	100:0	100	8.05	-	2.49	-	-	-
M4/MAPC1	20:80	20	0.26	0.45	0.15	1.45	0.16	0.10
	50:50		0.27	0.10	0.16	0.32	0.63	0.50
	80:20		0.24	0.07	0.14	0.23	2.50	0.63
	20:80	100	0.26	0.23	0.15	0.74	0.16	0.21
	50:50		0.33	0.26	0.19	0.84	0.63	0.23
	80:20		0.40	0.28	0.24	0.90	2.50	0.26
AC6/MAPC1	20:80	20	0.16	1.22	0.06	3.94	0.12	0.02
	50:50		0.18	0.45	0.07	1.45	0.50	0.05
	80:20		0.37	0.35	0.15	1.13	1.99	0.13
	20:80	100	0.21	0.33	0.08	1.06	0.12	0.08
	50:50		0.72	0.17	0.29	0.55	0.50	0.53
	80:20		0.63	0.13	0.25	0.42	1.99	0.61
M8/MAPC1	20:80	20	0.22	0.46	0.07	1.48	0.10	0.05
	50:50		3.82	0.56	1.18	1.81	0.39	0.65
	80:20		4.55	0.29	1.41	0.94	1.56	1.51
	20:80	100	0.53	0.31	0.16	1.00	0.10	0.16
	50:50		8.04	0.77	2.99	2.48	0.39	1.00
	80:20		6.63	0.24	2.05	0.77	1.56	2.65

^a^ Calculated according to Equation (3). ^b^ Measured by XRF and calculated according to Equation (1). ^c^ Calculated according to Equation (4). ^d^ Calculated according to Equation (5).

**Table 2 polymers-15-02169-t002:** Water (WCA) and diiodomethane (DCA) contact angles and water sliding angles (SAs) for the different flax fabrics modified with either the fluorinated or phosphonated monomers, or the combination of both.

Monomers Combination	Monomers Ratio (F/P)	Dose (kGy)	F-Content (wt%)	P-Content (wt%)	WCA (°)	DCA (°)	SA (°)
Pristine flax	-	0	0.00	0.00	-	-	-
MAPC1	0:100	20	0.00	0.22	-	-	-
	0:100	100	0.00	1.73	-	-	-
M4	100:0	20	0.15	0.00	138 ± 1	132 ± 2	43.5 ± 4.7
	100:0	100	0.22	0.00	130 ± 2	125 ± 5	90.0 ± 0.0
AC6	100:0	20	0.31	0.00	145 ± 4	128 ± 2	38.0 ± 1.6
	100:0	100	1.09	0.00	147 ± 3	134 ± 3	28.0 ± 3.7
M8	100:0	20	1.97	0.00	150 ± 2	133 ± 1	9.5 ± 1.3
	100:0	100	8.05	0.00	150 ± 1	136 ± 2	9.8 ± 1.3
M4/MAPC1	20:80	20	0.26	0.45	0 ± 0	0 ± 0	-
	50:50		0.27	0.10	124 ± 9	74 ± 10	>90
	80:20		0.24	0.07	133 ± 4	125 ± 2	>90
	20:80	100	0.26	0.23	137 ± 4	110 ± 13	>90
	50:50		0.33	0.26	140 ± 2	129 ± 3	46.5 ± 5.5
	80:20		0.40	0.28	142 ± 1	130 ± 2	37.0 ± 4.0
AC6/MAPC1	20:80	20	0.16	1.22	0 ± 0	0 ± 0	-
	50:50		0.18	0.45	129 ± 7	121 ± 3	45.0 ± 3.5
	80:20		0.37	0.35	145 ± 2	130 ± 4	35.0 ± 3.0
	20:80	100	0.21	0.33	94 ± 23	88 ± 13	>90
	50:50		0.72	0.17	135 ± 9	125 ± 8	42.0 ± 4.5
	80:20		0.63	0.13	142 ± 1	131 ± 3	31.0 ± 5.5
M8/MAPC1	20:80	20	0.22	0.46	0 ± 0	0 ± 0	-
	50:50		3.82	0.56	146 ± 2	134 ± 1	12.8 ± 2.8
	80:20		4.55	0.29	147 ± 2	131 ± 1	13.5 ± 1.5
	20:80	100	0.53	0.31	95 ± 3	62 ± 4	-
	50:50		8.04	0.77	151 ± 1	131 ± 3	11.3 ± 1.4
	80:20		6.63	0.24	149 ± 2	132 ± 2	13.5 ± 1.5

**Table 3 polymers-15-02169-t003:** Main data from PCFC for the flax fabric sample modified by fluorinated and/or phosphonated monomers.

Monomers Combination	Ratio (F:P)	Dose (kGy)	F-Content (wt%)	P-Content (wt%)	pHRR (W/g)	T-pHRR (°C)	THR (kJ/g)	Residue (wt%)
Pristine flax	-	0	0.00	0.00	230	370	9.0	11
MAPC1	0:100	20	0.00	0.22	131	312	6.8	21
	0:100	100	0.00	1.73	47	255	2.9	40
M4	100:0	20	0.15	0.00	199	367	9.7	13
	100:0	100	0.22	0.00	206	365	9.5	13
AC6	100:0	20	0.31	0.00	183	368	8.4	14
	100:0	100	1.09	0.00	210	363	9.6	11
M8	100:0	20	1.97	0.00	207	375	9.9	8
	100:0	100	8.05	0.00	210	364	9.1	8
M4/MAPC1	20:80	20	0.26	0.45	95	302	4.5	26
	50:50		0.27	0.10	139	323	6.7	17
	80:20		0.24	0.07	208	329	9.5	13
	20:80	100	0.26	0.23	121	312	5.6	22
	50:50		0.33	0.26	106	304	5.4	24
	80:20		0.40	0.28	122	302	6.9	18
AC6/MAPC1	20:80	20	0.16	1.22	46	271	2.8	37
	50:50		0.18	0.45	102	300	4.9	28
	80:20		0.37	0.35	106	302	5.3	24
	20:80	100	0.21	0.33	83	288	4.6	28
	50:50		0.72	0.17	157	306	7.9	16
	80:20		0.63	0.13	200	323	9.2	14
M8/MAPC1	20:80	20	0.22	0.46	84	288	4.4	27
	50:50		3.82	0.56	63	289	5.4	24
	80:20		4.55	0.29	85	298	8.1	11
	20:80	100	0.53	0.31	90	299	4.4	23
	50:50		8.04	0.77	53	279	6.1	28
	80:20		6.63	0.24	106	307	6.8	19

**Table 4 polymers-15-02169-t004:** Main data from cone calorimeter test for pristine and modified flax fabrics (100 kGy—M4/MAPC1 50/50).

	Sample Weight (g)	P-Content (wt%)	F-Content (wt%)	Cone Calorimetry				
				TTI (s)	pHRR (kW/m^2^)	THR * (kJ/g)	EHC * (kJ/g)	Residue * (wt%)
Pristine flax fabric	1.9 (1.8–1.9)	0.00	0.00	28 (25–31)	102 (98–105)	15.7 (15.1–16.4)	15.7 (15.1–16.4)	0
F-100 kGy-M4/MAPC1 (50:50)	2.2 (2.2–2.2)	0.77	0.49	14 (13–15)	98 (98–98)	10.7 (10.4–10.9)	12.5 (11.9–13.1)	15 (13–17)

* measured 150 s after the beginning of the test (after flame out).

## Data Availability

All data are reported in this article.

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
