# Peer review of "One-Step Multifunctionalization of Flax Fabrics for Simultaneous Flame-Retardant and Hydro-Oleophobic Properties Using Radiation-Induced Graft Polymerization"

_polymers, 2023, doi:10.3390/polym15092169_

Round 1

Reviewer 1 Report

This work describes a radiation-assisted surface functionalization of flax fibers with acrylate-based PFAS precursors. While the overall work is of good technical merit and conclusions support their findings, some suggestions below will improve the manuscript for publication.

1. The authors do not make it clear in the IR section how the actual chemistry of the acrylate radicals are covalently binding to the flax surface. What are the actual chemical linkages?  It's hard to conclude after washings if PFAS remains physio-absorbed or chemically modified, since the washings only involved non-fluorinated solvents which may not dissolve of PFAS monomers. 

2. Biggest issue is the content is simply too long for a journal article, even for a full paper. The 13 figures and more 4 tables can be easily consolidated into supporting information, leaving only relevant content in the main manuscript. 

3. While this work is comprehensive and new relative to other studies regarding improving textile properties, PFAS grafting onto surfaces has been extensively done before, so this work is an incremental contribution to the literature and is deemed average. 

4. It's not clear why this submission is going to a special issue in RAFT. There's no RAFT chemistry in this submission. 

Author Response

This work describes a radiation-assisted surface functionalization of flax fibers with acrylate-based PFAS precursors. While the overall work is of good technical merit and conclusions support their findings, some suggestions below will improve the manuscript for publication.

  1. The authors do not make it clear in the IR section how the actual chemistry of the acrylate radicals are covalently binding to the flax surface. What are the actual chemical linkages?  It's hard to conclude after washings if PFAS remains physio-absorbed or chemically modified, since the washings only involved non-fluorinated solvents which may not dissolve of PFAS monomers. 

Response:

The radiografting procedure is based on a first step where the fabrics are irradiated to form radicals in the flax structure followed by a second step where these fabrics are kept  into contact with the solution containing the monomer. Indeed, the radical species formed during the irradiation step initiate the copolymerization reaction of the (meth)acrylate monomers. In the absence of an irradiation step before the treatment of flax fabrics with the solution containing the fluorinated monomer, no trace of fluorine is detected at the end of the procedure after washing. Moreover, the various fluorinated monomers are soluble in THF which was used as the first washing solvent while the corresponding polymers are soluble in 2-butanone (MEK) used for the second washing. This was detailed in the experimental part regarding the grafting procedure: “The final step is the washing of the treated fabrics three times with THF and three times with MEK at room temperature for P(M4) and at 60°C for P(AC6) and P(M8) to remove unreacted monomers and free fluorinated polymer chains which were not covalently bonded to the flax structure.

  1. Biggest issue is the content is simply too long for a journal article, even for a full paper. The 13 figures and more 4 tables can be easily consolidated into supporting information, leaving only relevant content in the main manuscript. 

Response: Scheme 1 and Figures 2, 3 and 5 have been shifted to the Supporting Information file to become Scheme S1, Figures S1, S2 and S3, respectively. Thus, the revised manuscript contains 11 figures.

  1. While this work is comprehensive and new relative to other studies regarding improving textile properties, PFAS grafting onto surfaces has been extensively done before, so this work is an incremental contribution to the literature and is deemed average. 

Response: We respectfuly disagree with the reviewer since no work on simultaneous radiografting using a preirradiation procedure, of phosphorous methacrylate and fluorinated (meth)acrylate(s) onto flax fibers has already been reported in the literature. Moreover, another interest of this work is the study of the impact of the presence of both monomers on the properties induced by the other monomer.

  1. It's not clear why this submission is going to a special issue in RAFT. There's no RAFT chemistry in this submission. 

Response: This should be a mistake when I submitted the manuscript. However, this was an invitation.

Reviewer 2 Report

This manuscript has great innovative significance in investigating one-step multifunctionalization of flax fabrics for simultaneous flame retardant and hydro-oleophobic properties using radiation induced graft polymerization. The work can arouse wide interests of researchers in design and preparation of new functional materials. The manuscript is interesting. In my frank opinion, the manuscript should be deserved for its final publication in such high-level Journal. The main reasons are as follows:

1. At first, the English ABSTRACT should be revised, and a unified simple present tense should be used. 

2. The research significance and future work should be described in the final stage of the abstract.

3. Aims need to be concisely stated and added at the end of introduction. Not only what was done/investigated, but why.

4. The reference is a little outdated, please update it. As seen in introduction about “Superhydrophobic surfaces were inspired by lotus leaves, with a water contact angle greater than 150° and an ultra-low sliding angle (less than 10°). Indeed, the surface of lotus leaves displays self-cleaning and anti-contamination properties due to the presence of micro- and nanostructures that increase the roughness and reduce the droplet…”, such as:

[a] Tian Shi, Jingsong Liang, Xuewu Li, Chuanwei Zhang, Hejie Yang. Improving the Corrosion Resistance of Aluminum Alloy by Creating a Superhydrophobic Surface Structure through a Two-Step Process of Etching Followed by Polymer Modification. Polymers 2022, 14(21), 4509.

[b] Li Xuewu, Gao Rui, Huang Yanfei, Guo Weiling, Wang Wuping, Xu Keren, Xing Zhiguo, Wang Haidou. Effect of heat treatment temperature on structural and electrical properties of plasma sprayed KNMN coatings. Integrated Ferroelectrics 2023, 232:114-126.

In such reference, the above reference should be quoted and added for great correlation with that the effect of superhydrophobic surface modification technique is described in detail with excellent stability and durability.

5. Under normal conditions, in conclusion section, important conclusions should be more concisely and simplified elaborated point by point for brevity and prominence, such as a) … … b) … … c) … ….

6. And also in the last point future research work should be given in conclusion section.

7. Authors have not mentioned grafting parameters in section 2. Please provide the data and also the corresponding reason for selecting such data? OR refer to the literature that has been reported?

Author Response

Reviewer 2

This manuscript has great innovative significance in investigating one-step multifunctionalization of flax fabrics for simultaneous flame retardant and hydro-oleophobic properties using radiation induced graft polymerization. The work can arouse wide interests of researchers in design and preparation of new functional materials. The manuscript is interesting. In my frank opinion, the manuscript should be deserved for its final publication in such high-level Journal.

Response: We appreciate your laudatory comments.

The main reasons are as follows:

  1. At first, the English ABSTRACT should be revised, and a unified simple present tense should be used.

Response: Though this is a bit surprising from usually used in article, we have followed the reviewer’s suggestion and have corrected.

  1. The research significance and future work should be described in the final stage of the abstract.

Response: This has also been corrected.

  1. Aims need to be concisely stated and added at the end of introduction. Not only what was done/investigated, but why.

Response: This has been modified to match the reviewer’s comment.

  1. The reference is a little outdated, please update it. As seen in introduction about “Superhydrophobic surfaces were inspired by lotus leaves, with a water contact angle greater than 150° and an ultra-low sliding angle (less than 10°). Indeed, the surface of lotus leaves displays self-cleaning and anti-contamination properties due to the presence of micro- and nanostructures that increase the roughness and reduce the droplet…”, such as:

[a] Tian Shi, Jingsong Liang, Xuewu Li, Chuanwei Zhang, Hejie Yang. Improving the Corrosion Resistance of Aluminum Alloy by Creating a Superhydrophobic Surface Structure through a Two-Step Process of Etching Followed by Polymer Modification. Polymers 2022, 14(21), 4509.

 This reference reports a two-step process of etching procedure that led to superhydrophobic coatings. It has been inserted in the revision.

[b] Li Xuewu, Gao Rui, Huang Yanfei, Guo Weiling, Wang Wuping, Xu Keren, Xing Zhiguo, Wang Haidou. Effect of heat treatment temperature on structural and electrical properties of plasma sprayed KNMN coatings. Integrated Ferroelectrics 2023, 232:114-126.

This article is out of the scope (dealing with Mn-doped KNN, KNMN, piezoelectric ceramic coatings).

In such reference, the above reference should be quoted and added for great correlation with that the effect of superhydrophobic surface modification technique is described in detail with excellent stability and durability.

Response: Though these articles report specific processes from inorganic substrates (out of the scope of polymers), only the first one has been inserted in the revised manuscript.

  1. Under normal conditions, in conclusion section, important conclusions should be more concisely and simplified elaborated point by point for brevity and prominence, such as a) … … b) … … c) … ….

Response: We have modified the conclusion.

  1. And also in the last point future research work should be given in conclusion section.

Response: We thank the reviewer for this remark, this part has been added in the conclusion section: “Further to this work, the impact of the affinity of the fluorinated monomers with the reaction solvent should be better evaluated in order to control the localization of the grafting or the texturing of the polymer coating formed on the fibres. A study of the mechanical properties of such functionalized flax fabrics would also allow evaluating if the grafting induces a reinforcement or embrittlement of the fibers. It would also be particularly worth studying the washing resistance of the treatments. Indeed, the treatment developed here allows to graft covalently the flame retardant groups or the water-oleophobic agent and thus will be necessary to evaluate its resistance to washing in time.”

  1. Authors have not mentioned grafting parameters in section 2. Please provide the data and also the corresponding reason for selecting such data? OR refer to the literature that has been reported?

Response: We respectfully disagree with the reviewer. In previous works, we elaborated in detail the separate grafting of these monomers used in this present work and studied parameters such as irradiation dose, nature and concentration of both monomers, time and temperature of the reaction. As reported in the manuscript, the objective of this work is to investigate the simultaneous grafting of phosphonated and fluorinated monomers in one step in order to provide both the flame retardant and hydro-oleophobic properties to flax fibers, respectively. In addition, the monomer concentration ratios F/P has been examined as a parameter.

Reviewer 3 Report

1. The authors devote more space in 3.31 and 3.3.2 to the testing of phosphorus and fluorine content, what is the significance of this work for improving the functionality of linen fabrics?

2. the Grafting process is not clearly presented in the article "2. Grafting process", the author should add a simple flowchart to make the logic of the article clearer and thus more reader-friendly

3. what are the reasons for the difference in the contact angles and the sliding angles when the monomers ratio is different? This result should be described and explained by the author.

4. The author should give due consideration to the need to add other monomer ratios to the article to make the results more convincing.

5. In the case of grafted flax, the mechanism by which the flame resistance of the fabric is enhanced after treatment should be fully described in order to make the flame resistance of the fabric more convincing.

6. The author should explain what the calorimetry (PCFC)-HRR curves in Figure 10 mean and why they indicate that the fabric is flame retardant.

7. It is important to note that your manuscript needs to be carefully edited by a professional English editor, with particular attention to English grammar, spelling, sentence structural integrity and punctuation, so that the reader clearly understands the objectives and results of the study.

Author Response

  1. The authors devote more space in 3.31 and 3.3.2 to the testing of phosphorus and fluorine content, what is the significance of this work for improving the functionality of linen fabrics?

Response: Indeed, the significance of this work relies on i) proving that an improvement of fire retardancy and hydro-oleophobic of flax could be reached and ii)  assessing the role of the contents of phosphorated and fluorinated compounds as well as the P/F ratio on these functional properties.

  1. the Grafting process is not clearly presented in the article "2. Grafting process", the author should add a simple flowchart to make the logic of the article clearer and thus more reader-friendly

Response: The scheme of the grafting process has been added to the manuscript (page 6).

  1. what are the reasons for the difference in the contact angles and the sliding angles when the monomers ratio is different? This result should be described and explained by the author.

Response: The parameters which controls the contact angles are the fluorine content but also the F/P ratio. As explained in the manuscript page 22 : “These surface properties are controlled by the concentration of the fluorine element but also by the F/P monomer ratio in the treated fabric. Indeed, when this ratio is too low, the sample remains hydrophilic and oleophilic“. Indeed, a low F/P ratio corresponds to a high MAPC1 amount compared to that of the fluorinated monomer. MAPC1 is known to be hydrophilic due to its phosphonate group (good solubility of the monomer and of the polymer in water). Hence, when its concentration is too high about that of the fluorinated monomer, the hydrophilic nature of the multigrafting becomes preponderant. It is also understood that a more in-depth study could be envisaged to finely determine the limit value of this F/P ratio from which one would change from a hydrophilic character to a hydrophobic one.

  1. The author should give due consideration to the need to add other monomer ratios to the article to make the results more convincing.

Response: This study highlights the one-step grafting of two monomers by the pre-irradiation method. We chose only three concentration ratios for this experiment, but we believe that these were sufficient initially to demonstrate the effectiveness of the treatment in modifying the flame retardant and hydroolephobic properties of the flax. However, future work aims to study the effect of parameters such as the nature of the reaction solvent on the efficiency of the grafting but also to test other monomer concentration ratios to assess the impact on the fire bahavior and surface properties.

  1. In the case of grafted flax, the mechanism by which the flame resistance of the fabric is enhanced after treatment should be fully described in order to make the flame resistance of the fabric more convincing.

Response: A previous work (ref. 27 ) reports the grafting of flame retardancy of treated fabric. The mechanism has already been described in the manuscript in part 3 (page 26):

These results are attributed to the fact that the phosphonated group of MAPC1 units acts as a flame retardant. With the increase of temperature, this group decomposes causing the formation of phosphoric acid, which can induce a phosphorylation of the primary hydroxyl group of cellulose to form a phosphorus ester [23,49]. These esters catalyze the dehydration of cellulose at low temperature, leading to char formation [49]. Therefore, charring is assisted by the presence of phosphorus, leading to higher residue yield and lower THR but decreased thermal stability compared to pristine flax fabrics.

  1. The author should explain what the calorimetry (PCFC)-HRR curves in Figure 10 mean and why they indicate that the fabric is flame retardant.

Response: PCFC-HRR figures show that the increase of the grafted phosphorus rate in the treated fabrics leads to a direct decrease of the peak HRR and THR (the area under the HRR curve). This trend is very common for phosphorated flame retardants used to promote charring of a diverse selection of substrates according to the mechanism described in part 3 in this manuscript.

  1. It is important to note that your manuscript needs to be carefully edited by a professional English editor, with particular attention to English grammar, spelling, sentence structural integrity and punctuation, so that the reader clearly understands the objectives and results of the study.

Response: A careful checking has been done by an English spoken researcher of our Laboratory. He could bring many valuable improvements.

Round 2

Reviewer 1 Report

I'm fine with the revisions and this incremental advance to the area of irradiated graft polymers on flax fibers will be well received across a variety of applied fields.